# High Accuracy of Three-Dimensional Navigated Kirschner-Wire-Less Single-Step Pedicle Screw System (SSPSS) in Lumbar Fusions: Comparison of Intraoperatively Planned versus Final Screw Position

**DOI:** 10.3390/brainsci14090873

**Published:** 2024-08-29

**Authors:** Mateusz Bielecki, Blake I. Boadi, Yizhou Xie, Chibuikem A. Ikwuegbuenyi, Minaam Farooq, Jessica Berger, Alan Hernández-Hernández, Ibrahim Hussain, Roger Härtl

**Affiliations:** Weill Cornell Medicine—Department of Neurosurgery, NewYork-Presbyterian Och Spine Hospital, New York, NY 10034, USAdrahh9208@gmail.com (A.H.-H.);

**Keywords:** accuracy, minimally invasive, screw placement, navigation, 3D-NAV, pedicle screw, SSPSS, MIS, MISS

## Abstract

(1) Background: Our team has previously introduced the Single-Step Pedicle Screw System (SSPSS), which eliminates the need for K-wires, as a safe and effective method for percutaneous minimally invasive spine (MIS) pedicle screw placement. Despite this, there are ongoing concerns about the reliability and accuracy of screw placement in MIS procedures without traditional tools like K-wires and Jamshidi needles. To address these concerns, we evaluated the accuracy of the SSPSS workflow by comparing the planned intraoperative screw trajectories with the final screw positions. Traditionally, screw placement accuracy has been assessed by grading the final screw position using postoperative CT scans. (2) Methods: We conducted a retrospective review of patients who underwent lumbar interbody fusion, using intraoperative 3D navigation for screw placement. The planned screw trajectories were saved in the navigation system during each procedure, and postoperative CT scans were used to evaluate the implanted screws. Accuracy was assessed by comparing the Gertzbein and Robbins classification scores of the planned trajectories and the final screw positions. Accuracy was defined as a final screw position matching the classification of the planned trajectory. (3) Results: Out of 206 screws, 196 (95%) were accurately placed, with no recorded complications. (4) Conclusions: The SSPSS workflow, even without K-wires and other traditional instruments, facilitates accurate and reliable pedicle screw placement.

## 1. Introduction

With accurate and safe screw placement, short- and long-term complications such as the penetrance of the spinal canal or adjacent vessels, pseudoarthrosis, adjacent segment disease, and neural injury or irritation can be avoided [1,2,3]. Many surgeons have used intraoperative navigation for screw placement to ensure precision. Navigation provides remarkable soft-tissue and bone imaging quality that is superior to fluoroscopy [4,5].

Pedicle screw placement has traditionally required pedicle probing instruments such as Kirschner wires (K-wires) and Jamshidi needles. Some surgeons believe that such instruments are absolutely necessary for accurate and reproducible placement of pedicle screws. Our group, however, implemented an advanced spinal fusion technique, known as the navigated Single-Step Pedicle Screw System (SSPSS), in which the use of K-wires was eliminated for percutaneous pedicle screw insertion. Instead of a K-wire or Jamshidi needle, a navigated stylet was incorporated at the tip of the navigated screw inserter which allowed for docking on the superior articulating facet of the indicated level. This was followed by screw insertion over the stylet [6].

The navigated SSPSS workflow tackles many of the challenges associated with conventional pedicle screw placement. Vertebral anatomy varies across patients, and with navigation, surgeons can meticulously plan a screw trajectory and localize the preferred region to be instrumented efficiently. Without navigation, surgeons are required to consistently verify anatomy with fluoroscopy which can prolong a procedure. Jamshidi needles and K-wires are also tedious to work with and can cause nerve damage or vascular injury [7,8]. With docking, tapping, and screw insertion combined into one seamless procedure with the SSPSS device, the repetitive and laborious process of various instrument use and fluoroscopic confirmation is eliminated.

In our recently published study, the “totally navigated” SSPSS workflow produced a 90% grade 0 breach [5,6]. When we assessed the accuracy of implanted screws in our previously published study, we only assessed anatomical accuracy postoperatively with CT scans, as much of the existing literature has. We subsequently sought a more detailed approach to assessment and decided to compare planned intraoperative screw trajectories with final screw position using the Gertzbein and Robbins classification system [9]. A comparison between intended pedicle screw placement and final screw position offers surgeons a better evaluation of how closely their surgical execution matches their planning. If there are any major deviations between intended and final screw position, surgeons can study these errors and devise plans to avoid them in the future. Given our previously reported 90% grade 0 breach, we anticipated a high percentage of screws placed exactly as intraoperatively planned.

## 2. Materials and Methods

### 2.1. Patient Information

This study was a retrospective single-center study focused on consecutive patients receiving instrumented fusion for lumbar degenerative disc disease. Indications for surgery included spondylolisthesis with or without neurogenic claudication, foraminal stenosis, or facet arthropathy with degenerative dynamic instability. Figure 1 shows a sample case. Patients were included from January 2022 until August 2022. Those included were over 18 years of age and underwent a transforaminal lumbar interbody fusion (TLIF), anterior lumbar interbody fusion (ALIF), or lateral lumbar interbody fusion (LLIF) procedure with the navigable SSPSS workflow. Patients with previous lumbar spinal surgery, osteoporosis, and spine fractures were excluded. Informed consent was obtained from all included patients.

### 2.2. Surgical Approach

Following general endotracheal anesthesia set-up and neurophysiologic monitoring needles, patients were positioned in the prone position. Intraoperative electromyography (EMG) monitoring was operated by a neurophysiology specialist throughout the procedure to avoid nerve injury. Excess skin and fat were taped down. The Brainlab (Brainlab AG, Munich, Germany) reference array was then fixed into the iliac crest using Steinmann pins. An intraoperative CT scan was performed and registered with navigation software. A pedicle screw trajectory was subsequently planned and graded with the 3D navigation system wand and stored for subsequent analysis. The SSPSS device with navigation star was registered and calibrated after the navigation wand was used to identify the desired trajectory (Figure 2). A stylet at the tip of the screw insertion device was used to dock on the aspect of the superior articulating facet of the level to be engaged both ipsilaterally and contralaterally. With this stylet, the surgeon was able eliminate K-wires and other risky pedicle-probing instruments from the surgical workflow. With gentle malleting, the stylet was inserted in the bone, and the screw was fully engaged into the pedicle. Once the pedicle screw was safely inserted into the pedicle, the device with the stylet attached was retracted. All screws used were Viper Prime System Screws (DePuy Spine, Raynham, MA, USA). As cited in our previous study, the navigated SSPSS workflow can be briefly summarized in the following steps:A skin incision is marked using navigation guidance.Navigation is manually verified using the Brainlab pointer to identify and palate a transverse process at a distance from the reference array.The navigated screw with the screw driver is calibrated.After inserting the screw, the screws are test stimulated with an extended electrode probe. A threshold of 8 mA is used to consider screw repositioning.A final intraoperative CT is completed with the navigation reference in place in case of further instrument adjustment or decompression.The patient’s wound is generously irrigated and washed after meticulous hemostasis is performed. Osteo-stimulative bone graft is packed under the rod. Local anesthesia is used to infiltrate the muscle and the wound is closed [6].

### 2.3. Analysis

General patient data collected included patient age, sex, and surgical specific outcomes including fusion technique, levels fused, revisions, and complications. Planned screw trajectories were compared with final screw placement based on the Gertzbein and Robbins grading system (Figure 3) [9]. This classification system requires the user to grade the anatomical placement of a screw ranging from grade A, meaning fully within the pedicle without any breaches to grade E, demonstrating a breach outside of the pedicle cortex by more than 6 mm (Figure 4). The complete grading system described by Gertzbein and Robbins is as follows: Grade A: screw completely within the pedicle, Grade B: screw breaches the pedicle cortex by less than 2 mm, Grade C: screw breaches pedicle cortex by less than 4 mm, Grade D: screw breaches pedicle cortex by less than 6 mm, Grade E: screw breaches pedicle by greater than 6 mm [9]. Breaches can be cranial, lateral, medial, or caudal (Figure 4). Screw trajectories were measured intraoperatively using Brainlab software, and images and measurements were stored. Final screw insertion was graded on postoperative CT scans using the Vue Motion PACS system. Accuracy was defined as a screw that had the same classification when compared to the intraoperative planned screw (Figure 5). Grade discrepancies with breaches between planned and final screw position were considered misplaced screws. Screw placement was graded by two fellowship-trained spine surgeons independently and disagreements were resolved by discussion. All other relevant clinical and operative information were collected from electronic medical records.

## 3. Results

### 3.1. Patient and Surgery Characteristics

Between January and August 2022, 47 patients with lumbar interbody fusions were included in this analysis. Table 1 provides an overview of patient and surgery characteristics. The mean age of the cohort was 64.1 ± 13.3 years. A total of 24 (51.1%) were male. The mean BMI was 27.6 ± 5.1. Of all patients, 38 (80.9%) underwent a TLIF, and 5 (10.6%) underwent an XLIF, while the remaining 4 patients received an ALIF (2, 4.3%) and a PLIF (2, 4.3%). A total of 38 patients (80.9%) were fused at one level, 7 (14.9%) at two levels, and 2 (4.3%) at three levels. Spinal stenosis was the most prevalent diagnosis, affecting 34% of patients, followed by lumbar radiculopathy (23.4%) and spondylolisthesis (19.1%). Intraoperative blood loss was typically low, with 27.7% of patients losing less than 50 mL and 48.9% losing between 50 and 150 mL. The median duration of surgery was 182 min (IQR: 144–225), and the median hospital stay was 48 h (IQR: 24–108). There were no intraoperative or immediate postoperative reported complications in this cohort.

### 3.2. Screw Placement and Accuracy

Table 2 provides an overview of the number of screws per vertebral body. All levels operated received bilateral screws. In total, 206 screws were placed with 103 on both the right and left side. Most screws were placed at L5 (84 screws) followed by L4 (68 screws). A total of 2 screws were placed at L1, 4 at L2, 20 at L3, and 28 at S1. In total, 10 screws (4.9%) placed were classified as breaches. These screws were placed at L4 on the left side (four screws), L4 on the right side (two screws), L5 on the left side (three screws), and S1 on the right side (one screw). Table 3 demonstrates how the intended placement of these screws deviated from their final location according to the Gertzbein and Robbins classification. Eight screws were planned as “A” but were placed as a “B”. One screw was perioperatively planned as “A” and was placed as a “C”. Finally, one screw was planned as a “B” and was placed as an “A”. In total, 196/206 (95%) screws were placed accurately.

## 4. Discussion

We sought to assess the accuracy of the SSPSS workflow for screw placement in the lumbar spine based on comparisons between intended/planned screw placement and the final location of the screw following placement. Previous studies have focused primarily on the anatomical accuracy of screw placement and screw breach evasion using only postoperative imaging. With the use of a navigation system, we were able to store our intraoperatively planned screw trajectories and compare these trajectories to CT imaging of the implanted screw. A total of 95% of pedicle screws inserted were inserted as “A”. Some screws intraoperatively planned as “grade B” were inserted as “grade A”. Such placement would not be defined as accurate with our definition. However, these screws were still implanted completely within the pedicle.

The impetus for our described SSPSS workflow was eliminating tedious traditional MIS methods and instrumentation such as K-wires. The use of K-wires carries risk. They can bend, causing difficulty with the passage of cannulated instruments over them. They can also penetrate the vertebral body, usually in osteoporotic patients, causing vascular or visceral injury or migrating into the spinal canal and causing infection. These complications can increase patient morbidity, operative time, radiation exposure, and surgeon frustration [10,11,12]. The navigated SSPSS workflow also addresses variations in patient anatomy that make screw placement difficult. With preoperative planning, surgeons can study patient anatomy in high detail and choose the best screws and implants for each patient. Intraoperatively, surgeons can also observe patient anatomy in real time. This starkly contrasts with repetitive fluoroscopy shots for anatomical understanding and localization. Spine surgeons can be actively engaged throughout the procedure without interruption with real-time navigation. Repetitive cannulation in multiple steps is also required for conventional screw placement. The literature has shown that the traditional, non-navigated method for the placement of pedicle screws is more likely to require intraoperative correction when compared to navigated screws. With intraoperative CT, poorly placed screws can be identified and re-positioned before the conclusion of the operation. This diminishes the need for revision surgeries. A human cadaver study confirmed that CT improved pedicle screw placement accuracy by up to 87%. [13] A meta-analysis found that out of 8539 screws placed, the perforation risk for navigation was 6% compared to 15% for free-hand placement without CT [14]. As the accuracy of screw placement increases with the assistance of intraoperative CT, the amount of radiation delivered to surgical staff also decreases [14,15,16]. With the navigated SSPSS system, pedicle screw placement becomes seamless. Docking, malleting, and screw insertion become integrated into a continuous routine, reducing unnecessary mental and physical demands on the surgeon.

No revision surgeries and neurologic or vascular complications were reported in our first described use of the SSPSS workflow, which led us to conclude that outcomes of fusions with the technique are favorable and traditional instrumentation like K-wires are not necessary for safety and accuracy [6]. In this study, we report a high accuracy of screws placed as planned with the same SSPSS workflow, once again demonstrating no need for stand-alone K-wires, Jamshidi needles, or other pedicle probing instruments in fusions for lumbar degenerative disc disease. Additionally, we observed that intraoperative blood loss was low, with 48.9% losing between 50 and 150 mL, and the mean surgical time was 182 min. These results demonstrate the efficiency and effectiveness of the SSPSS. The streamlined nature of this system likely contributed to reduced intraoperative bleeding and shorter surgical times compared to traditional multi-step systems.

While the SSPSS workflow does incorporate a navigated stylet that does serve somewhat of a similar function to the K-wire, its small size and ease of use make the stylet quite different and superior. With a navigation system, the surgeon can easily monitor the style in real time as it is docked and engaged into the patient’s bone. This greatly reduces the risk of visceral and vascular injury or unintended migration that K-wires pose. The navigation system we used also undoubtedly accounted for the high accuracy we observed between intraoperatively planned and postoperative screws. Previous studies have confirmed that non-navigated screws were more likely to require intraoperative correction than navigated screws [6].

Miller et al. assessed the accuracy of implanted pedicle screws in the thoracic and lumbosacral spine based on a planned trajectory [17]. Of 240 screws placed, the mean angular difference between the planned trajectory and implanted screw was 2.17° ± 2.20° on axial images and 2.16° ± 2.24° on sagittal images. These angular differences are small and are comparable to the high placement accuracy we achieved in our study. Ganguly et al. also reviewed the difference between 59 planned and postprocedural pedicle screw locations and observed a small mean difference of 2.8 mm; however, K-wires were used in this study [18]. To our knowledge, this is the first study to compare intraoperatively planned screw positions with final placements in evaluating pedicle screw placement accuracy using the SPSS workflow without K-wires [19,20,21,22].

Fujita et al. reconstructed a 3D model from pre-operative CT data using MySpine^®^ system (accessible via the MySpine web planner). They designed a pre-operative surgical plan for pedicle screw placement which included screw diameter, length, and direction in transverse and sagittal angles. Using this model as a guide, pedicle screws were placed. The authors then compared the 3D models with postoperative CT scans. Of all previous studies reviewed, this method of screw accuracy assessment is most similar to ours. However, there are differences. With the use of the Brainlab navigation system, we were able to plan and place pedicle screws during the operation. Fujita et al. planned pedicle screw insertion preoperatively. Fujita et al. also compared planned and final screw positions with angular deviations, while our study used the Gertzbein and Robbins grading system to evaluate differences. An important similarity between both studies is the patient-specific pedicle screw planning afforded by both the MySpine (accessible via the MySpine web planner) and Brainlab systems (Munich, Germany) [23].

A surgical challenge of utilizing SSPSS is the potential for motion changes by surgeons during pedicle screw insertion to impact the recognition of intraoperative imaging. One scenario where we often encounter an intraoperative change in screw trajectory, differing from the preoperative plan, is when there is skiving of the stylet extending out of the SPSS due to a steep angle at the SAP/transverse process junction. In such cases, the screw may deviate laterally and fail to capture the pedicle. To overcome this challenge, we identify a new starting point that does not match the preplanned trajectory and adjust the trajectory to maximize pedicle capture. We use a navigated awl-tip tap, which has a sharper tip and can either follow the initial starting point or establish a new one. If this method is ineffective, we use the new starting point with the stylet. These adjustments help ensure accurate screw placement despite the challenges posed by intraoperative motion.

This study is not without its limitations. First, we recognize that the sample size for our study is small. As we continue to use the SSPSS workflow in lumbar fusions, we will continue to assess the accuracy of the technique with technological advancements like AR and robotics in more patients. Second, the single-arm design does not include a comparative analysis with the conventional free-hand technique. This limits our ability to definitively conclude that the observed benefits are superior to those achieved with traditional methods. Future studies should incorporate comparative, randomized controlled trials to provide a more robust assessment of the efficacy and advantages of the single-step pedicle screw system over conventional techniques. Third, we only used one method for accuracy assessment. We believe, however, that our use of the Gertzbein and Robbins grading system is a valid and reliable method to assess accuracy. However, we acknowledge the intrinsic differences in screw classification. Specifically, deviation from the initial plan should consider the entry point, the tip of the screw, and angulation rather than simply differentiating between GRS A, B, and C screws. While we focused on GRS due to its clinical significance and widespread acceptance, we recognize that future studies assessing accuracy should incorporate more precise measurements of screw placement accuracy. This includes evaluating deviations in the entry point, the tip of the screw, and angulation. Additionally, considering the ratio of pedicle diameter to screw diameter is crucial for a comprehensive assessment. Future research comparing intraoperative trajectory planning and postoperative imaging with the SSPSS workflow should therefore expand beyond classification systems like GRS and include numerical deviations for a more precise evaluation of screw placement accuracy.

## 5. Conclusions

Overall, our study demonstrates a high accuracy of pedicle screw placement when compared with intraoperatively planned trajectories for lumbar interbody fusions. K-wires and other traditional pedicle probing instruments are not needed for the accurate implantation of pedicle screws if the SSPSS workflow is employed.

## Figures and Tables

**Figure 1 brainsci-14-00873-f001:**
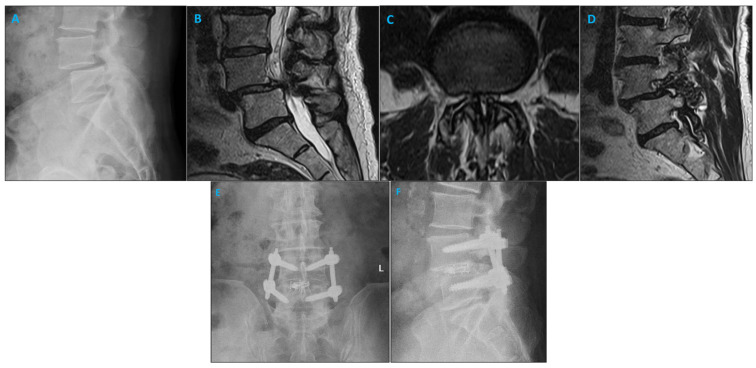
Imaging studies of a patient with degenerative spondylolisthesis who underwent minimally invasive TLIF and pedicle screw placement using the SPSS. (**A**) Lateral X-ray demonstrating L4/5 spondylolisthesis. (**B**) Sagittal MRI illustrating the left L4/5 foramen. (**C**) Axial MRI of the L4/5 level shows narrowing of the lumbar spinal canal. (**D**) Sagittal MRI illustrating the right L4/5 foramen. (**E**) Coronal postoperative radiograph after L4/5 minimally invasive TLIF. (**F**) Sagittal postoperative radiograph after L4/5 minimally invasive TLIF.

**Figure 2 brainsci-14-00873-f002:**
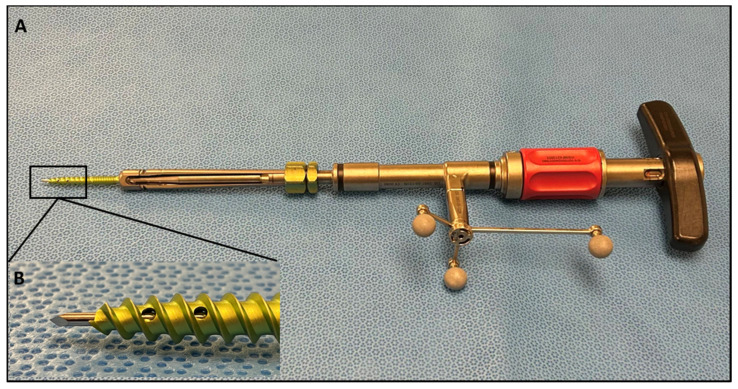
(**A**) SSPSS navigated screw inserter with spheres attached, forming the navigation star. The navigation star is crucial for navigation registration and intraoperative anatomical localization. (**B**) Close-up view of the screw inserter showing the navigated stylet.

**Figure 3 brainsci-14-00873-f003:**
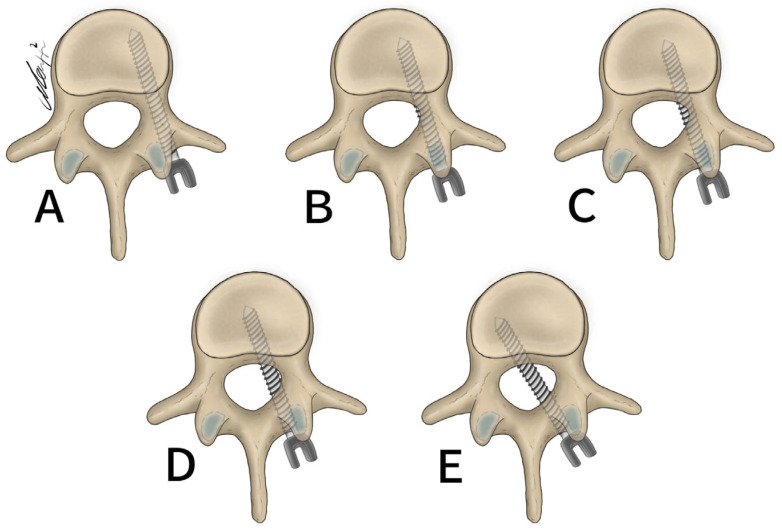
Gertzbein and Robbins classification system. (**A**) 0 mm breach. (**B**) <2 mm breach. (**C**) <4 mm breach. (**D**) <6 mm breach. (**E**) >6 mm breach.

**Figure 4 brainsci-14-00873-f004:**
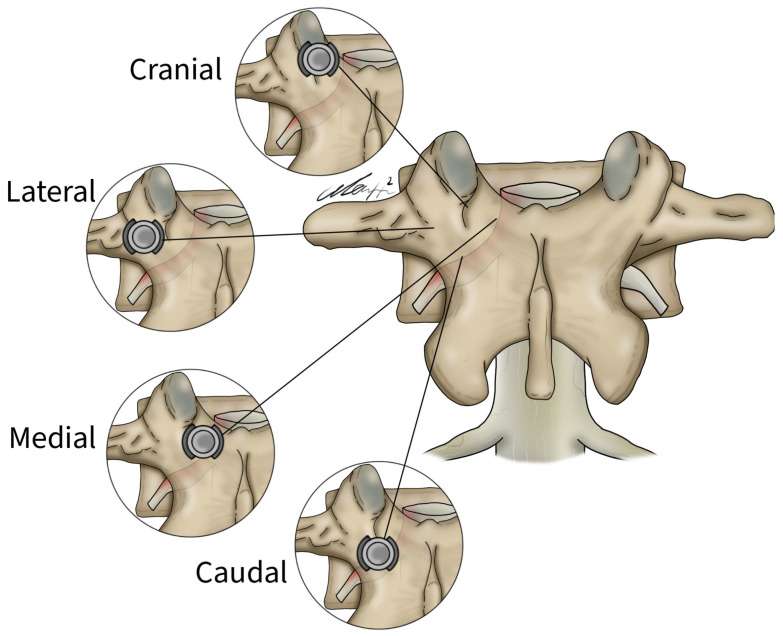
Lateral, medial, cranial, and caudal screw breaches demonstrated within the vertebral body.

**Figure 5 brainsci-14-00873-f005:**
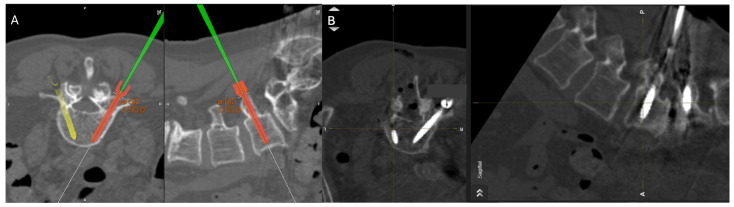
Percutaneous pedicle screw insertion intraoperative imaging. (**A**) Grade A screw trajectory planned at the right L5 with navigation wand. (**B**) Final pedicle screw inserted within planned Grade A trajectory at the right L5 pedicle.

**Table 1 brainsci-14-00873-t001:** Patient and surgery characteristics.

Characteristic	Frequency (*n* = 47)
Age (years)	64.1 ± 13.3
Sex	
Male	24 (51.1%)
Female	23 (49.9%)
BMI (kg/m^2^)	27.6 ± 5.1
Fusion technique	
ALIF	2 (4.3%)
PLIF	2 (4.3%)
TLIF	38 (80.9%)
XLIF	5 (10.6%)
Number of levels fused	
1	38 (80.9%)
2	7 (14.9%)
3	2 (4.3%)
Primary Diagnosis	
Spinal stenosis	16 (34)
Lumbar radiculopathy	11 (23.4)
Spondylolisthesis	9 (19.1)
DDD	6 (12.8)
Others	5 (10.6)
Revision during follow-up	2 (4.3%)
Adjacent segment disease	2
Blood Loss (*n*, %)	
<50 mL	13 (27.7)
50–150 mL	23 (48.9)
>150 mL	11 (23.4)
Duration of surgery (minutes), median (IQR)	182 (144–225)
Length of hospital stay (hours), median (IQR)	48 (24–108)

DDD, degenerative disc disease; Others (revision, revision and extension, spondylosis, closed unstable burst fracture).

**Table 2 brainsci-14-00873-t002:** Number of screws inserted per vertebral levels.

Vertebral Body	Screw Position		Total
	Right	Left	
L1	1	1	2
L2	2	2	4
L3	10	10	20
L4	34 (2) *	34 (4) *	68
L5	42	42 (3) *	84
S1	14 (1) *	14	28
Total	103	103	206

* Number of screws that changed from the planned position.

**Table 3 brainsci-14-00873-t003:** Comparison between intraoperatively planned screws and final screw position using the Gertzbein and Robbins classification system.

	Inserted Screws				
Planned		A	B	C	Total
	A	195	8	1	204
	B	1	1	0	2
	C	0	0	0	0
	Total	196	9	1	206

## Data Availability

The data presented here are available upon request from the corresponding author. They are not publicly available because of ethical concerns.

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
