# Peer review of "High Accuracy of Three-Dimensional Navigated Kirschner-Wire-Less Single-Step Pedicle Screw System (SSPSS) in Lumbar Fusions: Comparison of Intraoperatively Planned versus Final Screw Position"

_brainsci, 2024, doi:10.3390/brainsci14090873_

Round 1

Reviewer 1 Report

Comments and Suggestions for Authors

Authors present a retrospective single center study on 47 patients who underwent lumbar interbody fusion to investigate accuracy and safety of 3D Navigated K-wire-less Single Step Pedicle 2 Screw System (SSPSS) with comparison of intraoperatively planned vs. final screw position. Authors claim the study to be novel, since K-wires or Jamshidi needles were not used. However, this looks as a simple navigated procedure with a navigated screwdriver, which is nothing new. Stylet which was attached at the tip of the screw is unfortunately not visualized in a Figure. Figure 1. depicts however a standard navigated percutaneous screw, with array attached to the screwdriver. This manuscript has one intrinsic fail - normally, all screws are planned as GRS A screws; therefore, deviation from the initial plan cannot be B screw vs. A. screw, but deviation of the entry point, of the tip of the screw and deviation of angulation. In practice, deviations almost always occurs, however A, B and C screws are usually clinically acceptable, and therefore not revised following iCT. I suggest for authors to provide a more exact measurment of screw placement accuracy, for example deviation of the entry point, tip and angulation. Ratio of pedicle diameter and screw diameter also needs to be taken into consideration. 

Author Response

Ref. No.: Manuscript ID ERD-2023-ST-0085 Titled “High Accuracy of 3D Navigated K-wire-less Single Step Pedicle Screw System (SSPSS) in Lumbar Fusions: Comparison of Intraoperatively Planned versus Final Screw Position

Dear Editors and Reviewers,

Thank you for your time and efforts. Please find our revised manuscript attached, in which we implemented and addressed the reviewers' requests and concerns. We have made the utmost effort to address all reviewers’ comments thoroughly, and the following are point-by-point responses to each reviewer.

Comments for the Author. 

Please see our responses below in blue. 

Reviewers' comments:

Reviewer 1.

Authors present a retrospective single center study on 47 patients who underwent lumbar interbody fusion to investigate accuracy and safety of 3D Navigated K-wire-less Single Step Pedicle 2 Screw System (SSPSS) with comparison of intraoperatively planned vs. final screw position. Authors claim the study to be novel, since K-wires or Jamshidi needles were not used . However, this looks as a simple navigated procedure with a navigated screwdriver, which is nothing new.

Response:

Thank you for your comments. While we agree that the absence of k-wires or Jamshidi needles for navigation is not a novel concept, the methodology of our study presents a novel aspect. Specifically, we compared the intraoperatively planned screw positions to the final screw placements, analyzing the differences between them. This comparative analysis has not been previously reported in studies evaluating navigated screw placement. We have adjusted the manuscript to emphasize this unique aspect of our methodology for clarity. See lines 248 – 250

Stylet which was attached at the tip of the screw is unfortunately not visualized in a Figure. Figure 1 . depicts however a standard navigated percutaneous screw, with array attached to the screwdriver.

Response:

Thank you for your comment. We have added a figure 1B which shows the stylet attached to the tip of the screw

This manuscript has one intrinsic fail - normally, all screws are planned as GRS A screws; therefore, deviation from the initial plan cannot be B screw vs. A. screw,  but deviation of the entry point, of the tip of the screw and deviation of angulation. In practice, deviations almost always occurs, however A, B and C screws are usually clinically acceptable, and therefore not revised following iCT. I suggest for authors to provide a more exact measurment of screw placement accuracy, for example deviation of the entry point, tip and angulation. Ratio of pedicle diameter and screw diameter also needs to be taken into consideration.

Response:

Thank you for your comment. We agree with this point and have added this as a limitation to the study in lines 258 – 271. See below:

Third, we only used one method for accuracy assessment. We believe that using the Gertzbein Robbins grading system is a valid and reliable method to assess accuracy. However, we acknowledge the intrinsic differences in screw classification, specifically that deviation from the initial plan should consider the entry point, the tip of the screw, and angulation rather than simply differentiating between GRS A, B, and C screws. While we focused on GRS due to its clinical significance and widespread acceptance, we recognize that future studies assessing accuracy should incorporate more exact measurements of screw placement accuracy, as you have mentioned. This includes evaluating deviations in the entry point, the tip of the screw, and angulation. Additionally, considering the ratio of pedicle diameter to screw diameter is crucial for a comprehensive assessment. Future research comparing intraoperative trajectory planning and postoperative imaging with the SSPSS workflow should therefore expand beyond classification systems like GRS and include numerical deviations for a more precise evaluation of screw placement accuracy.

Reviewer 2.

Many thanks for giving me an opportunity regarding the review this article.

Title: High Accuracy of 3D Navigated K-wire-less Single Step Pedicle Screw System (SSPSS) in Lumbar Fusions: Comparison of Intraoperatively Planned versus Final Screw Position

Outline: This study aims to evaluate the accuracy of 3D Navigated K-wire-less Single Step Pedicle Screw System (SSPSS) in Lumbar Fusions. From this study, the authors observed 196/206 (95%) of screws were placed accurately. No complications were recorded. Based on this result, the authors described the navigable SSPSS workflow allows for accurate and reliable placement of pedicle screws despite the elimination of K-wires and other conventional instruments in the surgical workflow. It is very interesting concept and sufficiently has academic values. However, several concerns were presented.

Comment:

  1. Further methods need to be supplemented. Are the preoperative CT/MR essential for working 3D Navigated K-wire-less Single Step Pedicle Screw System? I think preoperative planning needs to be illustrated

Response

Thank you for your comment. Preoperative CT/MR scans are not strictly essential for the 3D Navigated K-wire-less Single Step Pedicle Screw System, as intraoperative CT is utilized. We rely on intraoperative CT because imaging fusion can differ between prone and supine positions, which may affect preoperative planning accuracy. Preoperative CT/MR scans are used to identify patient pathology and select the appropriate procedure, but intraoperative imaging ensures more accurate alignment and placement, given the positional differences.

  1. Please illustrated disease entity such as degenerative lumbar disease et al. It need to be added because the authors present it's inclusion criteria (age more than 18 years)

Response

Thank you for your comment. This was already in the manuscript in the methods section, lines 71 – 73. However, we have added the various distributions of diagnoses the patient presented within the result section, lines 141 – 143. Furthermore, we have added a case example with corresponding imaging of a patient with degenerative spondylolisthesis who underwent a minimally invasive TLIF and pedicle screw placement using the SPSS.

“Indications for surgery included spondylolisthesis with or without neurogenic claudication, foraminal stenosis, or facet arthropathy with degenerative dynamic instability. Patients were included from January 2022 until August 2022. Those included were over 18 years of age…”

  1. During the procedures for SSPSS, motion changes by surgeons during pedicle screws insertion may be influenced on recognition of intraoperative imaging. Please discuss the author's experiences for surgical challenge for SSPSS.

Response

Thank you for your comment. We agree with your point regarding the potential for motion changes by surgeons during pedicle screw insertion to impact the recognition of intraoperative imaging. One scenario where we often encounter an intraoperative change in screw trajectory, differing from the preoperative plan, is when there is skiving of the stylet extending out of the SPSS due to a steep angle at the SAP/transverse process junction. In such cases, the screw may deviate laterally and fail to capture the pedicle.

To overcome this challenge, we must identify a new starting point that does not match the preplanned trajectory and then adjust the trajectory to maximize pedicle capture. One method we use is the navigated awl tip tap, which has a sharper tip and can either follow the initial starting point or establish a new one. If this method is ineffective, we use the new starting point with the stylet. These adjustments ensure accurate screw placement despite the challenges posed by intraoperative motion.

We added this point to the discussion lines 221 – 231

  1. Please add the operative outcomes including intra-operative bleeding, surgical time and discuss the their results regarding the utility of SSPSS.

Response

Thank you for your comment. We agree that operative outcomes, including intra-operative bleeding surgical time, and discussing the result of their utility of SSPSS will be beneficial in improving the manuscript. This has been added to the results lines 143 – 145 and Table 1. Furthermore, we have added this to the discussion section of the manuscript, lines 215 -219

  1. Please add the limitation regarding single-arm study (not, comparative study with conventional free-hand technique) in the Discussion section.

Response

Thank you for your comment. We agree that the study's single-arm nature is a limitation. We have added this to the limitation. Lines 245 – 258 See below.

“…a notable limitation of our study is its single-arm design, which does not include a comparative analysis with the conventional free-hand technique. This limits our ability to definitively conclude that the observed benefits are superior to those achieved with traditional methods. Future studies should incorporate comparative, randomized controlled trials to provide a more robust assessment of the efficacy and advantages of the single-step pedicle screw system over conventional techniques.”

Handling Editor

  1. There are some sentences similar to former publications (mainly in the

Abstract). To avoid any misunderstanding or trouble, we would like to ask you

to rephrase all the marked places to avoid duplication or self-plagiarism

problem. Attached please find an repetition report of your paper.

Response

Thank you for your comments. We have rephrased all the marked places in the abstract to avoid duplication or self-plagiarism. See below

(1) Background: Our team has previously introduced the Single Step Pedicle Screw System (SSPSS), which eliminates the need for K-wires, as a safe and effective method for percutaneous minimally invasive spine (MIS) pedicle screw placement. Despite this, there are ongoing concerns about the reliability and accuracy of screw placement in MIS procedures without traditional tools like K-wires and Jamshidi needles. To address these concerns, we evaluated the accuracy of the SSPSS workflow by comparing the planned intraoperative screw trajectories with the final screw positions. Traditionally, screw placement accuracy has been assessed by grading the final screw position using post-operative CT scans. (2) Methods: We conducted a retrospective review of patients who underwent lumbar interbody fusion, using intraoperative 3D navigation for screw placement. The planned screw trajectories were saved in the navigation system during each procedure, and post-operative CT scans were used to evaluate the implanted screws. Accuracy was assessed by comparing the Gertzbein and Robbins classification scores of the planned trajectories and the final screw positions. Accuracy was defined as a final screw position matching the classification of the planned trajectory. (3) Results: Out of 206 screws, 196 (95%) were accurately placed, with no recorded complications. (4) Conclusion: The SSPSS workflow, even without K-wires and other traditional instruments, facilitates accurate and reliable pedicle screw placement.

  1. Please add *back matters* to your manuscript, including Author

Contributions, Funding, Institutional Review Board Statement, Informed

Consent Statement, Data Availability Statement, Acknowledgments, and

Conflicts of Interest. For more detailed information, please see the attached

template.

Response:

Thank you for your comment. We have added this to the manuscript. See below:

Author Contributions: Conceptualization, Ibrahim Hussain and Roger Härtl; methodology, Mateusz Bielecki, Blake I. Boadi, Yixhou Xie, Ibrahim Hussain, and Roger Härtl; software, Ibrahim Hussain and Roger Härtl; validation, Ibrahim Hussain and Roger Härtl; formal analysis, Mateusz Bielecki, Yixhou Xie, and Chibuikem A. Ikwuegbuenyi; investigation, Mateusz Bielecki, Blake I. Boadi, Yixhou Xie, Ibrahim Hussain, and Roger Härtl; resources, Roger Härtl; data curation, Blake I. Boadi, and Jessica Berger; writing—original draft preparation, Mateusz Bielecki, Blake I. Boadi, Yixhou Xie, and Ibrahim Hussain; writing—review and editing, Mateusz Bielecki, Blake I. Boadi, Yixhou Xie, Chibuikem A. Ikwuegbuenyi, Minaam Farooq, Jessica Berger, Alan Hernández, Ibrahim Hussain, and Roger Härtl MD1; visualization, Alan Hernández; supervision, Ibrahim Hussain and Roger Härtl; project administration, Ibrahim Hussain and Roger Härtl; funding acquisition, Roger Härtl. All authors have read and agreed to the published version of the manuscript.

Funding: This research received no external funding

Institutional Review Board Statement: This study was performed in accordance with the ethical standards of the Declaration of Helsinki (as revised in 2013) and was approved by the Weill Cornell Medicine (WCM) Institutional Review Board (IRB) (approval number: 1912021199) and individual consent for this retrospective analysis was waived.

Informed Consent Statement: Patient consent was waived due to the retrospective nature of this study. The extracted data included clinical data only and did not include any personally identifiable information. Therefore, the need for informed consent was waived.

Data Availability Statement: The data presented here are available upon request from the corresponding author. They are not publicly available because of ethical concerns.

Acknowledgments: N/A

Conflicts of Interest: Roger Hartl declares consulting work for DePuy Synthes, Brainlab, and Ulrich. Roger Hartl reports a financial relationship with Zimmer Biomet and Real Spine. No other author declares any financial interests or personal relationships

Please use the version of your manuscript found at the above link for your

revisions. Please make all requested changes to the Word file attached; we

have applied our journal's layout style.

Reviewer 2 Report

Comments and Suggestions for Authors

Many thanks for giving me an opportunity regarding the review this article.

Title: High Accuracy of 3D Navigated K-wire-less Single Step Pedicle Screw System (SSPSS) in Lumbar Fusions: Comparison of Intraoperatively Planned versus Final Screw Position

Outline: This study aims to evaluate the accuracy of 3D Navigated K-wire-less Single Step Pedicle Screw System (SSPSS) in Lumbar Fusions. From this study, the authors observed 196/206 (95%) of screws were placed accurately. No complications were recorded. Based on this result, the authors described the navigable SSPSS workflow allows for accurate and reliable placement of pedicle screws despite the elimination of K-wires and other conventional instruments in the surgical workflow. It is very interesting concept and sufficiently has academic values. However, several concerns were presented.

Comment:

1. Further methods need to be supplemented. Are the preoperative CT/MR essential for working 3D Navigated K-wire-less Single Step Pedicle Screw System? I think preoperative planning needs to be illustrated

2. Please illustrated disease entity such as degenerative lumbar disease et al. It need to be added because the authors present it's inclusion criteria (age more than 18 years)

3. During the procedures for SSPSS, motion changes by surgeons during pedicle screws insertion may be influenced on recognition of intraoperative imaging. Please discuss the author's experiences for surgical challenge for SSPSS.

4. Please add the operative outcomes including intra-operative bleeding, surgical time and discuss the their results regarding the utility of SSPSS.

5. Please add the limitation regarding single-arm study (not, comparative study with conventional free-hand technique) in the Discussion section.

Author Response

(The authors gave the same response as above.)

Round 2

Reviewer 1 Report

Comments and Suggestions for Authors

Authors have sufficiently responded to remarks. 

Reviewer 2 Report

Comments and Suggestions for Authors

no more comments